# Consequences of COVID-19 for the Pancreas

**DOI:** 10.3390/ijms23020864

**Published:** 2022-01-13

**Authors:** Urszula Abramczyk, Maciej Nowaczyński, Adam Słomczyński, Piotr Wojnicz, Piotr Zatyka, Aleksandra Kuzan

**Affiliations:** 1Department of Medical Biochemistry, Wroclaw Medical University, 50-368 Wroclaw, Poland; aleksandra.kuzan@umw.edu.pl; 2Faculty of Medicine, Wroclaw Medical University, 50-368 Wroclaw, Poland; maciej.nowaczynski@student.umw.edu.pl (M.N.); adam.slomczynski@student.umw.edu.pl (A.S.); piotr.wojnicz@student.umw.edu.pl (P.W.); piotr.zatyka@student.umw.edu.pl (P.Z.)

**Keywords:** COVID-19, pancreas, SARS-CoV-2, diabetes

## Abstract

Although coronavirus disease 2019 (COVID-19)-related major health consequences involve the lungs, a growing body of evidence indicates that COVID-19 is not inert to the pancreas either. This review presents a summary of the molecular mechanisms involved in the development of pancreatic dysfunction during the course of COVID-19, the comparison of the effects of non-severe acute respiratory syndrome coronavirus 2 (SARS-CoV-2) on pancreatic function, and a summary of how drugs used in COVID-19 treatment may affect this organ. It appears that diabetes is not only a condition that predisposes a patient to suffer from more severe COVID-19, but it may also develop as a consequence of infection with this virus. Some SARS-CoV-2 inpatients experience acute pancreatitis due to direct infection of the tissue with the virus or due to systemic multiple organ dysfunction syndrome (MODS) accompanied by elevated levels of amylase and lipase. There are also reports that reveal a relationship between the development and treatment of pancreatic cancer and SARS-CoV-2 infection. It has been postulated that evaluation of pancreatic function should be increased in post-COVID-19 patients, both adults and children.

## 1. Effects of Severe Acute Respiratory Syndrome-Related Coronavirus (SARS-CoV) and Middle East Respiratory Syndrome-Related Coronavirus (MERS-CoV) on the Pancreas

Coronaviruses are enveloped, single- and positive-stranded RNA viruses that infect birds and mammals. In humans, coronaviruses cause respiratory tract infection, usually the common cold, but they can also cause severe respiratory illness including severe acute respiratory syndrome (SARS) and Middle East respiratory syndrome (MERS), caused by severe acute respiratory syndrome-related coronavirus (SARS-CoV) and Middle East respiratory syndrome-related coronavirus (MERS-CoV), respectively [1]. Coronaviruses tend to cause epidemics and even pandemics. The first coronavirus pandemic was the SARS outbreak in 2002–2003 [2]. With the experience gained during the SARS pandemic, it was possible to more quickly identify subsequent outbreaks of the MERS epidemic in 2012 [3]. The pathomechanism of both viruses is very similar—they even both use transmembrane protease serine 2 (TMPRSS2), except SARS-CoV uses angiotensin-converting enzyme 2 (ACE2) as its receptor, whereas MERS uses dipeptidyl peptidase-4 (DPP4) [4,5]. Moreover, there is a difference in terms of the severity and frequency of symptoms, which was observed in MERS patients as more frequent hospitalization in the intensive care unit (ICU) compared to SARS patients [2] (Table 1). Diabetes was one of the significant and independent predictors for developing severe SARS-CoV and MERS-CoV [6,7,8]. In MERS, no viral antigen was detected in any tissue other than pneumocytes [7], despite multiple organ dysfunction syndrome in critically ill patients. In SARS-CoV, the presence of the virus was detected not only in respiratory epithelial cells, but also in small intestinal and colonic epithelial cells, in which it also revealed replication features [9]. It is known that the ACE2 receptor is also present in tissues such as the heart, kidney, and pancreas [8,9]. According to some authors, the presence of the receptor is sufficient for tissue entry and pathogenic activity, although other researchers do not support this thesis [9,10]. Yang et al. were some of the first researchers who hypothesized that SARS coronavirus enters islets using ACE2 as its receptor and damages islets causing acute diabetes [8]. Yang’s study revealed that SARS-CoV had a much higher affinity for pancreatic islet cells than for pancreatic exocrine cells, which was consistent with the hyperglycemia observed in some patients and rarely reported acute pancreatitis (AP) [8]. Furthermore, insulin-dependent diabetes mellitus (IDDM) and high fasting blood glucose values were observed in some inpatients [8]. A 3-year follow-up revealed that both abnormalities were transient, which may be indicative of only temporary damage to the pancreatic islets [8]. However, another reason (different from that given by Young et al.) for high fasting blood glucose value in patients may result from increased stress hormones release. Cortisol, catecholamines, growth hormone, and glucagon, which are released during infection, fever, and trauma, can lead to hyperglycemia to the same degree as SARS-CoV can [11]. No information was found in the literature about a direct impact of the MERS virus on the pancreas or on glycemia during or after infection. This may be due to an insufficiently detailed analysis of the available data during previous studies that oscillated primarily, for laboratory tests, between complete blood count (CBC), lactate dehydrogenase (LDH), urea, and creatinine analysis. A summary of SARS-CoV, MERS, and SARS-CoV-2 is shown in Table 1.

In 2019, a new coronavirus named SARS-CoV-2 was identified, causing COVID-19. This virus has many characteristics that are analogous to SARS-CoV, for example, ACE2 is also used as its receptor [12]. Patients with diabetes are among those with the most severe forms of COVID-19 and related mortality; insights from recent experience can guide future management [17], particularly for the consequences on the pancreas. As the COVID-19 pandemic has been ongoing for nearly two years, this study aims to collect data concerning the impact of SARS-CoV-2 on the pancreas and analyze them to estimate the future health consequences of COVID-19 in populations.

## 2. Pancreatic Damage during Diabetes Mellitus and COVID-19

Pancreas tissue damage may cause to the lack of control over normal blood glucose levels in the body. Type 1 diabetes (T1D) is caused by insulin deficiency due to βcell dysfunction of immunologic or idiopathic cause. In contrast, β pancreatic cells in type 2 diabetes (T2D) become depleted over time due to compensatory insulin secretion caused by insulin resistance. There is also type 3 diabetes (T3D), which is described as diabetes associated with the development of Alzheimer’s disease [18]. It should not be confused with type 3c (pancreatogenic) diabetes, which relates to the exocrine and digestive functions of the pancreas. The issue concerning the impairing effect of hyperglycemia (glucotoxicity) on the secretory function of the islets of Langerhans has also been increasingly raised. In addition to endocrine dysfunction, some diabetic patients may also develop moderate exocrine pancreatic insufficiency (EPI), in which pancreatic enzyme secretion is impaired. EPI can be observed in almost all patients with type 3c (pancreatogenic) diabetes (secondary to pancreatic pathology), whereas the prevalence of this dysfunction in patients with T1D or T2D is 40% and 27%, respectively [19].

With the ongoing SARS-CoV-2 pandemic, patients with reduced normal pancreatic function are at high risk for COVID-19 requiring hospitalization. In particular, elevated blood glucose levels in patient with and without diabetes makes them at high risk of mortality [20]. Hyperglycemia impairs the immune response (e.g., by reducing the activity of macrophages and polymorphonuclear leukocytes), which in addition influences the excessive cytokine response, and thus has a strong proinflammatory effect.

The receptors for ACE2, which are also present in the pancreas, are a target of SARS-CoV-2 in the body, which may result in acute failure of both the islets of Langerhans and exocrine cells [15]. Infection-induced, transient β cell dysfunction may cause an uncontrolled hyperglycemic state, especially in patients whose pancreas is already affected by diabetes mellitus. Persistent hyperglycemia usually predisposes to severe COVID-19 and to viral infection complicated by secondary infections. The aforementioned risk can be found in T1D, T2D, and gestational diabetes mellitus (GDM). In T2D patients, the much more frequent coexistence of other risk factors such as atherosclerosis, hypertension, and obesity should be taken into consideration, which usually implies a worse prognosis for the course of COVID-19 [21,22]. In GDM, SARS-CoV-2 infection not only increases the risk of more severe course of the disease in a patient, but may also result in diabetic fetopathy or, in more advanced pregnancies, increase the risk of future pathologies involving glucose metabolism (such as T2D) in a child [23].

## 3. Pancreatic Damage in Patients without Pre-Existing Diabetes Infected with SARS-CoV-2

It has been postulated that, either by direct invasion of pancreatic cells by the virus or by indirect mechanisms described below, SARS-CoV-2 has a destructive effect on the pancreas and can lead to insulin deficiency and development of T1D [24].

If the hypothesis that SARS-CoV-2 infection causes hyperglycemia is true, increased statistics of new T1D cases should be observed. Indeed, there are publications that describe such a phenomenon. For instance, Unsworth et al. and Kamrath et al. describe an increase in new-onset T1D in children during the COVID-19 pandemic [16,25]. Although pancreatic β cell damage induced transient hyperglycemia in SARS-CoV, it is still unclear whether β cell damage is transient or permanent in SARS-CoV-2 [22]. This information appears to be of great importance because COVID-19 in children is frequently considered “harmless”. Therefore, it is reasonable to sensitize parents to the fact that the consequences of COVID-19 may be potentially dangerous for their children.

Below you will find the proposed molecular mechanisms that may participate in pancreatic damage that causes carbohydrate metabolism disorders.

## 4. Etiology Associated with ACE2, TMPRSS2, and Na+/H+ Exchanger

As previously mentioned, SARS-CoV infection of host cells is facilitated by ACE2, but also by the transmembrane protease serine 2 (TMPRSS2) and other host cell proteases such as cathepsin L (CTSL) [13].

ACE2 is an enzyme that is expressed to varying degrees in most cells of the human body [14,26,27]. This enzyme catalyzes the conversion of angiotensin II to angiotensin 1–7, taking part in the maintenance of body homeostasis by influencing the regulation of blood pressure and water–electrolyte balance through the renin–angiotensin–aldosterone (RAA) system [28]. Moreover, ACE2/angiotensin (1–7) stimulates insulin secretion, reduces insulin resistance, and increases pancreatic βcell survival [27,28].

In addition to the key role it plays in maintaining body homeostasis, ACE2 is now also the best-studied target for SARS-CoV-2 S glycoprotein, enabling infection of host cells [27,29]. ACE2 in the pancreas is expressed mainly within the pericytes of pancreatic microvessels and to a lesser extent on the surface of the islets of Langerhans, including pancreatic β cells [30]. SARS-CoV-2 shows 10–20 times more activity against ACE2 than SARS-CoV, which significantly increases the infectivity of SARS-CoV-2 [31,32]. Furthermore, studies indicate that SARS-CoV may also downregulate ACE2 expression in cells. This causes an imbalance between ACE and ACE2, consequently leading to blood pressure disorders and systemic inflammation [27,33,34]. Due to the 79% genetic similarity between SARS-CoV and SARS-CoV-2 [35], it is speculated that ACE2 expression may also be downregulated during SARS-CoV-2 infection, causing i.a. MODS observed in COVID-19 [27].

During cell infection by SARS-CoV-2, in addition to the role played by ACE2, it is also appropriate to consider the significant pathogenic role of TMPRSS2 that is necessary for the preparation of S glycoprotein by its cleavage, thereby enabling fusion of the virus with the host cell [36,37]. The S1 and S2 domains can be distinguished in the SARS-CoV-2 S glycoprotein. The S1 domain is involved in binding to the ACE2 receptor and then TMPRSS2 intersects with the S protein, including at the boundary of the S1 and S2 domains and within the S2 domain, which enables the virus–cell fusion [38,39]. According to studies, TMPRSS2 expression is significantly increased in obese patients, which may contribute to the poorer prognosis that is observed during COVID-19 in this patient group [40]. Moreover, obese patients are frequently already burdened with problems such as insulin resistance at baseline, while the presence of ACE2 and TMPRSS2 within the pancreas as a binding site for SARS-CoV-2 may exacerbate insulin resistance causing problems in terms of diabetes management in COVID-19 patients.

There are also other mechanisms by which COVID-19 may affect the development of hyperglycemia. It is reported that the virus may also affect the glucose regulation through the Na+/H+ exchanger and lactate pathways. The mechanism is that angiotensin II, which accumulates during infection, contributes to insulin resistance and—by activating the Na+/H+ exchanger in the pancreas—it leads to hypoxia and extracellular acidification, which, through the accumulation of calcium and sodium ions in the cells and the production of reactive oxygen species, damages pancreatic tissue [41]. Simultaneously, the concentration of lactate increases, which in COVID-19 infection is intensively released, among other things, from adipose tissue, and then monocarboxylate transporters transport lactate and H+ ion inward in the cell, which increases Na+/H+ exchanger activation, further disrupting pancreatic homeostasis [41].

## 5. The Etiology Associated with a Systemic Proinflammatory Environment, Immune System Aggression, and Production of Novel Autoantigens

A broad spectrum of proinflammatory cytokines, such as IL-2, IL-6, IL-7, IL-8, interferon-γ, and Tumor Necrosis Factor α (TNF-α), is released during, in particular severe, COVID-19 infection [42,43,44]. Based on current studies, it is reasonable to suspect that these cytokines are released in response to the binding of the virus to ACE2 receptors that are also located in the pancreas [9,42]. The cause of pancreatic damage during COVID-19 is the cytokine storm that plays a key role in this case, because in both acute pancreatitis (AP) and severe COVID-19, elevated levels of the aforementioned interleukins are associated with the severity of these both disease entities. Particular attention should be paid to IL-6, because it is suspected to play a key role in the pathogenesis of AP as well as acute respiratory distress syndrome (ARDS) that is the most common and most severe clinical manifestation of COVID-19. In COVID-19-induced ARDS, IL-6 levels are correlated with disease-related mortality [45,46,47]. At the same time, high IL-6 levels correlate with an increased risk of developing severe pancreatitis [48,49].

The production of neutralizing antibodies is also an important response of the body in the course of COVID-19 [50,51,52]. It has been observed that early seroconversion and very high antibody titers occur in patients with severe SARS-CoV-2 infection [53,54]. The available literature details a mechanism called antibody-dependent enhancement (ADE), which is associated with a pathological response of the immune system [53]. ADE exploits the existence of FcRS receptors located on various cells of the immune system, for example, macrophages and B lymphocytes [53]. This relationship may lead to a likely bypass of the classical viral infection pathway by ACE2, and virus–antibody complexes may stimulate macrophages to overproduce cytokines including significant IL-6 [53,55].

Molecular mimicry may be also one of potential causes of pancreatic cell damage [56]. There are similarities in the protein structure of the virus and β-pancreatic cells, which may induce cross-reactivity and lead to autoimmunity [56]. Furthermore, viral infection may also lead to increased cytokine secretion by surrounding dendritic cells and activation of naive T cells in genetically predisposed individuals [56].

## 6. Pancreatitis in COVID-19

Although the impact of the discussed coronavirus-induced disease on exocrine function is not fully understood, available literature is not able to unambiguously determine whether the tissue damage leading to AP occurs as a result of direct SARS-CoV-2 infection [57] or as a result of systemic MODS with increased levels of amylase and lipase [42]. Liu et al.’s study involving 121 COVID-19 patients with a mean age of 57 years and a variable course of infection proved above-normal levels of amylase and lipase in 1–2% of patients with moderate COVID-19 infection and in 17% of patients with severe COVID-19 infection. This may support the hypothesis that SARS-CoV-2-induced disease has a destructive effect not only on the endocrine portion of this gland, but also on the exocrine one [15].

However, elevated levels of pancreatic enzymes in question do not have to mean the destruction of pancreatic cells—after all, such a situation may occur during kidney failure or diarrhea in the course of COVID-19. Furthermore, there remains the question of the effect of drugs administered during SARS-CoV-2 infection on changes in pancreatic function [42], discussed further in this article.

According to the International Association of Pancreatology (IAP) and the American Pancreatic Association (APA), the diagnosis of AP is based on meeting two out of three of the following criteria: clinical (epigastric pain), laboratory (serum amylase or lipase > 3 × upper limit of normal), and/or imaging criteria (computed tomography, magnetic resonance imaging, ultrasound) [58]. Pancreatic lipase is considered as a potential marker of SARS-CoV-2 severity with concomitant AP. In Hemant Goyal et al.’s study, as many as 11.7% out of 756 COVID-19 patients had hyperlipidemia and they were three times more likely to have severe COVID-19 [59]. Those with higher lipase levels—17% out of 83 patients—required hospitalization [60]. However, it is difficult to distinguish whether these patients required hospitalization for severe systemic COVID-19 infection or for pancreatitis in the course of COVID-19 infection.

AP in the course of COVID-19 was analyzed in different age groups; however, some studies only involve children [61]. Compared to pancreatic islet cells, cells of the exocrine pancreatic ducts are more abundant in ACE2 and TMPRSS2 that are necessary for the virus to penetrate the cell [62]. Infection of these cells may be one of the causes of AP [63]. Infections, both bacterial and viral, are one of the causes of AP. The definitive mechanism of how viral infections affect pancreatic cells is not known; however, a study by Maria K Smatti et al. found that there is infection of pancreatic islet cells and replication of the virus within them, ultimately resulting in autoimmune reactions that eventually affect both diabetes and AP in a negative way [64]. For non-SARS-CoV-2 patients, the etiology of AP is known and confirmed in most cases, although 69% of those undergoing infection do not have definite etiology of AP while meeting the AP-Atlanta criteria for diagnosis [65].

Hegyi et al. show the mechanism of MODS formation during COVID-19 infection and AP [66]. This is lipotoxicity, involving an interstitial increase in pancreatic lipase levels, which leads to the breakdown of triacylglycerols contained in adipose tissue cells and the release of unsaturated fatty acids. These in turn exert a toxic effect on mitochondria causing the release of cytokines, which results in a cytokine storm.

There is also a hypothesis, which claims that AP can develop because of blood circulatory centralization resulting from uncontrolled cytokine storm created by SARS-CoV-2 infection [67]. There exist reports that say that pancreatic ischemia may be the cause of different degrees of acute pancreatitis [68,69]. This statement can be supported by the reports that state that pancreatic blood reperfusion inhibits the development of AP and accelerate pancreas recovery [70].

Another mechanism of developing AP during COVID-19 may be a coagulation cascade activation caused by active inflammatory process due to SARS-CoV-2 infection [71]. The ongoing inflammatory process causes not only hemostasis imbalance for blood clotting, but it also leads to intensification of coagulation by removing epithelial cell protein C receptor (EPCR) from epithelial by the means of inflammatory mediators and thrombin [71]. This means that both processes intensify each other. Simultaneously, it was proved that COVID-19 predisposes patients to venous thromboembolism resulting from excessive inflammation, platelet activation, and endothelial dysfunction [72]. It is also important to notice that AP is inherently connected with a coagulation cascade activation, increased fibrinolysis and, hence, higher level of D-dimers [73]. Acute pancreatitis severity may depend on hemostasis imbalance; local coagulation results in mild AP whereas, in more severe AP cases, the imbalance may lead to development of disseminated intravascular coagulation (DIC) [74]. These observations have been supported by the results of experimental studies showing that the inhibition of coagulation reduces the development of AP [75,76,77] and exhibits therapeutic effect in this disease [78,79]. Additionally it is worth noticing that infection-related hyperglycemia has powerful inflammation-promoting effects on the organism (especially when organism is under stress), thus increasing the number of inflammatory mediators [74]. Unfortunately, it is impossible to decide which process is dominant in causing AP in COVID-19 patients: local inflammation caused by SARS-CoV-2 or systemic hemostasis imbalance.

Clinical reports on low molecular weight heparin (LMWH) treatment in AP seem to emphasize a more significant role of hemostasis imbalance in causing AP [74,80,81]. Heparin is extremely significant in the treatment of COVID19 patients due to its properties, mainly its similarity to heparan sulphate, which appears in a respiratory tract, its interactions with SARS-CoV-2 S protein, leading to viral adhesion inhibiting to the cell membrane [82], and its anti-inflammatory effects. Thanks to these properties, heparin may not only show its therapeutic effect as the anticoagulant, but also its protective role in acute pancreatitis or respiratory inflammations [83,84,85].

## 7. Drugs Used against SARS-CoV-2 Infection (Glucocorticoids, Lopinavir, Ritonavir, Remedesivir, Interferon-β1 (IFN-β1), and Azithromycin) Induce Pancreatic β Cell Damage

Statistical analyses revealed a significantly higher incidence of AP with the concomitant systemic use of glucocorticosteroids (GCS) [86]. In one study analyzing the development of drug-induced AP, dexamethasone, was classified as type IB—there was one case report in which administration of this drug-induced AP occurred; however, other causes of pancreatitis such as alcohol consumption could not be excluded [87]. Other GCS such as hydrocortisone, prednisone, and prednisolone were used in patients with mild to moderate AP; however, they cannot be classified into any group because they are frequently used together with other drugs that cause AP [86,87]. However, it has been determined that GCS independently increase the risk of AP, and patients with residual AP risk factors during GCS treatment should be more monitored for the development of AP [23]. Javier A. Cienfuegos et al. additionally observed that one of mechanisms of AP formation in COVID-19 patients may be GCS administered at the time of admission to the ICU with severe respiratory failure [88]. Because GCS were used in severe COVID-19 cases, it is difficult to say what true reason for AP was—either a severe course of COVID-19 or GCS application or both.

GCS are used in the treatment of many diseases due to their immunosuppressive and anti-inflammatory nature. They induce diabetes in previously healthy patients as well as significantly exacerbate diabetes in diabetic patients [89,90]. Diabetes develops in these patients likely due to pancreatic β cell dysfunction, decreased insulin secretion, and increased insulin resistance in other tissues, which may depend on the timing and the dose of GCS used [89,91]. Long-acting or intermediate-acting insulin alone or combined with short-acting insulin should be used during the treatment [90]. At the same time, no advantage was found over the use of oral hypoglycemics [92]. Certainly, patients after long-term GCS therapy will need further observation for diabetes.

Lopinavir/ritonavir was classified in the previously mentioned study as a type IV drug—medications reported with little information [87]. Both drugs are included in the group of antiretrovirals that act as protease inhibitors, and they are primarily used for HIV infection. Although Lopinavir is an active drug, it is not used alone. There have been reports about the occurrence of AP during the use of protease inhibitors in question, which is also described in the Summary of Product Characteristics (SmPC) of products approved by Committee for Medicinal Products for Human Use (CHMP). It has been proved that the use of lopinavir/ritonavir causes hyperglycemia [93,94].

Remdesivir is an adenosine analogue with antiviral activity. There are single reports about the occurrence of pancreatitis as a result of the use of the aforementioned medication [95,96]. At the same time, it should be noted that other nucleoside-derivative drugs may cause pancreatitis [97].

The current state of knowledge does not clearly indicate the therapeutic benefit of interferon-β in the treatment of COVID-19 patients [98,99]. To date, only single cases suggesting induction of pancreatitis by interferon-β have been reported. Based on this, Badalov et al. classified interferon into type III [87].

There are few reports about the development of AP due to the use of azithromycin [100]. In the previously mentioned study by Badalov et al., two macrolide antibiotics were classified as type II and III. Unfortunately, there are no direct data concerning azithromycin. Interestingly, there were cases of patients with concomitant symptoms of AP and viral pneumonia caused by SARS-CoV-2 who were treated with azithromycin, which resulted in complete resolution of symptoms for both conditions [96,101]. Based on available data, the risk of azithromycin-induced AP is low.

There is no clear evidence that azithromycin affects blood glucose levels in humans. However, it is known for its prokinetic effects, which may be helpful in patients who suffer from diabetic gastroparesis [102]). The incidence of hypo- and hyperglycemic episodes was not proved to be significant for azithromycin [103]; however, the risk of dysglycemia is emphasized [94]. In the SmPC, where azithromycin is the main ingredient, it is not possible to establish a causal relationship between the occurrence of pancreatitis and taking medications (Zithromax) based on the available data. In contrast, glycemic disturbances were not indicated as side effects (Zithromax) [104].

Hydroxychloroquine has been extensively promoted for COVID-19 due to its anti-inflammatory and antiviral action; yet, the use of this agent in diabetes deserves particular attention for its documented hypoglycemic action, and its benefit on COVID-19 is controversial, although there is large usage [105].

Table 2 shows a comparison of the side effects of medications in question.

## 8. COVID-19, Pancreas, and Glycation

In T2D diabetics, oxidative stress leading to pancreatic damage may be stimulated by, among other things, the intense glycation that accompanies hyperglycemia [24]. Glycation is a non-enzymatic process involving reducing sugar and amino groups of proteins, which contributes to the formation of advanced glycation end products (AGEs). These products have significantly altered biochemical properties relative to the substrates, including proteins that have altered conformation, increased rigidity, resistance to proteolysis, etc. [106,107].

Part of the pathomechanism involved in facilitating coronavirus infection in diabetics may be due to glycation of ACE2 and SARS-CoV-2 spike protein [108,109].

An interesting hypothesis is that COVID-19 has a worse prognosis in patients with intense glycation, and thus high tissue AGE content. Glycated hemoglobin (HbA1c) is a commonly used diagnostic tool that estimates intensity of glycation. The parameter is not only a marker of long-term persistent hyperglycemia, but an active participant in immune processes, as HbA1c levels are associated with NK cell activity [110].

Zhang et al.’s retrospective cohort study concerning COVID-19 patients revealed that glycated hemoglobin correlates negatively with saturation (SaO2) and positively with C-reactive protein (CRP), erythrocyte sedimentation rate (ESR), and fibrinogen (Fbg). It was concluded that determination of HbA1c levels may be helpful in assessing inflammation, hypercoagulability, and prognosis of COVID-19 patients [111].

According to the meta-analysis by Chen et al. (2020), Hba1c levels were slightly higher in patients with severe COVID-19 compared to patients with mild COVID-19; however, this correlation was not statistically significant. However, it is of great importance to note that only two studies analyzing HbA1c in COVID-19 patients were included in this analysis because only these studies were available in May 2020 [112].

Glycation plays its physiological effects not only directly by changing the properties of various proteins, but also indirectly through various receptors. RAGE is the most common receptor for AGEs. Binding of RAGE to its ligands activates a proinflammatory response primarily by mitogen-activated protein kinase (MAPK) and nuclear factor κβ (NFκβ) pathways. This interaction was proved to be significant in the pathogenesis of cancer, diabetes mellitus, and other inflammatory disorders [113]. RAGE was found to be expressed in the pancreas, and S100P-derived RAGE antagonistic peptide (RAP) reduces pancreatic tumor growth and metastasis [113]. The implications of this fact may also apply to the etiology and treatment of COVID-19. It has been postulated that targeting RAGE by various antagonists of this receptor may inhibit damage to various organs including the pancreas [114].

## 9. COVID-19 vs. Pancreatic Cancer

Immunosuppression as a treatment effect, elevated cytokine levels, altered expression of receptors for SARS-CoV-2, and a prothrombotic state in patients with various types of cancer may exacerbate the effects of COVID-19 [115].

Focusing on pancreatic cancer, it can be observed that the pathomechanism of both diseases—COVID-19 and tumorigenesis in the pancreas—overlap in several molecular mechanisms. As mentioned above, SARS-CoV-2 infection of host cells is facilitated by ACE-2, TMPRSS2, and CTSL. Cathepsin L is upregulated in a wide variety of cancers, including pancreatic adenocarcinoma [13]. TMPRSS2 upregulation in pancreatic cancers is moderate, whereas ACE-2 is overexpressed in some cancers, including pancreatic carcinomas [115]. Interestingly, ACE2 upregulation seems to be associated with favorable survival in pancreatic cancer [116], and it is known that SARS-CoV-2 reduces ACE2 expression [22]. Furthermore, the above-mentioned RAGE may also participate in both pancreatic cancer development and SARS-CoV-2 infection. RAGE facilitates neutrophil extracellular trap (NET) formation in pancreatic cancer [117]. In conclusion, pancreatic cancer predisposes to an increased risk of COVID-19 and its more severe course, and coronavirus infection may contribute to pancreatic cancer.

It also seems important how the COVID-19 epidemic has affected the treatment of patients with pancreatic cancer of SARS-CoV-2-independent etiology. According to the study by Pergolini et al., care of patients with pancreatic cancer can be disrupted or delayed, particularly in the context of treatment selection, postoperative course, and outpatient care [118].

A separate issue is how patients after pancreatoduodenectomy respond to SARS-CoV-2 infection. A case series reported by Bacalbasa reveal that patients who develop SARS-CoV-2 infection postoperatively require re-admission in the ICU and a longer hospital stay; however, these infections are not fatal [119]. Although the analysis was performed on single cases, it is concluded that these results are an argument to perform elective oncological surgeries [119].

There are also reports that chemotherapy in pancreatic cancer patients who become ill between treatment series can be successfully completed after a complete cure of the infection [120]. Guidelines for, e.g., prioritization and treatment regimens regarding pancreatic cancer treatment in the era of the pandemic, are developed and described, for example, by Catanese et al. or Jones et al. [121,122].

## 10. Conclusions

Evidence shows that SARS-CoV-2 infection contributes to damage within the pancreas. The mechanisms that are involved in this include but are not limited to direct cytopathic effect of SARS-CoV-2 replication and systemic and local inflammatory response [123]. At the current state of knowledge, it is certain that the virus attacks the endocrine portion of the pancreas as well as, to a much lesser extent, the exocrine portion. It has been shown that a bidirectional relationship between COVID-19 and diabetes exists; indeed, diabetes is associated with COVID-19 severity and mortality but, at the same time, patients with COVID-19 have shown new onset of diabetes [124]. SARS-CoV-2 virus infection not only directly affects glycemic levels, but also exacerbates already existing hyperglycemia through its negative impact on the functional competence of the islets of Langerhans. It cannot be excluded that the real cause of exocrine dysfunction of this gland is the negative effect of the drugs used for treatment of the infection. As the pandemic progresses, special attention should be given to the evaluation of chronic and acute pancreatic diseases, including pancreatic cancer, so that faster diagnosis enables faster implementation of treatment.

## Figures and Tables

**Table 1 ijms-23-00864-t001:** The summary of characteristics of SARS and MERS coronaviruses. Dipeptidyl peptidase-4 (DPP4), transmembrane protease serine 2 (TMPRSS2), hospitalization in the intensive care unit (ICU), and cathepsin L (CTSL).

Comparison of Virus Characteristics			
Compared Characteristic	MERS-CoV	SARS-CoV	SARS-CoV-2
Receptor	DPP-4 [4], TMPRSS2 [7]	ACE2 [4] TMPRSS2 [5] CTSL [5]	ACE2 [12] TMPRSS2 [13] CTSL [13]
TMPRSS2	Essential for virus–cell fusion [5]	Essential for virus–cell fusion [5]	Essential for virus–cell fusion [13]
Cell under attack	Pneumocytes, activated leukocytes, liver and prostate, kidney [4,7]	Pneumocytes, small intestinal and colonic epithelial cells, arterial and venous endothelium, smooth muscle, macrophages [4,9]	Pneumocytes, kidney, gastrointestinal system, bladder cells [14]
Hospitalization in the ICU	Frequent [2]	Less frequent [2]	Frequent [15]
Acute Pancreatitis	No data	Single cases [8]	Single cases [15]
Hyperglycemia	No data	Transient [8]	Transient [16]

**Table 2 ijms-23-00864-t002:** Side effects of medications used in SARS-CoV-2 infection in the area of pancreatic effects and hyperglycemia.

	Side Effect	
Drug	Hyperglycemia	Pancreatitis
Glucocorticosteroids	Present	Increased risk
Lopinavir/Ritonavir	Present	Few cases
Remdesivir	No data	Few cases
Interferon-β	No data	Few cases
Azithromycin	Not present	Likely

## Data Availability

Not applicable.

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
