# Peer review of "Consequences of COVID-19 for the Pancreas"

_ijms, 2022, doi:10.3390/ijms23020864_

Round 1
Reviewer 1 Report
In this review authors satisfactorily described the possible implications of covid-19 with the pancreas.
I have a few comments listed below:
Which is the reference for lines 45-46? Did you mean "transient" diabetes requiring insulin or T1D?
At line 52 authors discussed the absence of DPP4 receptor on pancreatic cells. Can they comment about pancreatic expression of DPP4, identifyed as a functional receptor or ligand for a variety of different molecules?
At line 249 authors described isophane insulin as the insulin to be used in case of GCS however, in several countries this type of insulin has been discontinuated.
Reviewer 2 Report
Manuscript ID ijms-1499836
Title: Consequences of COVID-19 for the pancreas
Authors: Abramczyk et al.
The manuscript entitled “Consequences of COVID-19 for the pancreas” seems to be interesting. However, it must be pointed out that the manuscript contains some deficiencies and errors which should be removed before its acceptance for publication.
List of errors and deficiencies:
- All abbreviations should be presented in their full name at the point where they appear for the first time in the abstract and repeated in the body of the manuscript (for example see line 8, 29, 51, 320, 321). On the other hand, the abundance of abbreviations makes the text difficult to understand (for example, line 75, EPI).
- Line 32-33. The authors stated that “Diabetes was one of the significant independent risk factors for developing SARS and MERS [5–7]”. This statement is unclear and misleading. SARS means only severe acute respiratory syndrome without specifying its cause. The authors should introduce the abbreviation SARS-CoV at line 26 and use it throughout the following section of the manuscript. The sentence .in lines 32-33 should be modified.
- Line 40. The authors have written “Young et al. was of the first…”. Young et al. means “Young and co-authors” and for this reason the authors should write “were”.
- Line 45-48. The authors stated that “Furthermore, insulin-dependent diabetes mellitus (IDDM) and high fasting blood glucose values were observed in some inpatients. A 3-year follow-up revealed that both abnormalities were transient, which may be indicative of only temporary damage to the pancreatic islets [7]”. This concept may be correct. There is, however, another possible reason of these effects. The authors should remember that trauma, infection, fever, hypoxia, and stress lead to an increased release of stress hormones such as cortisol, catecholamines, growth hormone and glucagon, and these hormones lead to hyperglycemia. This information should be included in the manuscript with appropriate references.
- Table 1. Table 1 should also include data on SARS-CoV-2. Name of presented parameters should be presented in the first column. In addition, tables must be understandable without references to the body of the manuscript. All abbreviations used in Tables should be presented in full name in table headings or in table captions. In addition, the authors should present references for the data presented in the Table.
- Lines 9, 60. “COVID-19” means coronavirus disease 2019 and for this reason the form COVID-19 disease is incorrect.
- There is a total mess in references. References must be numbered in order of appearance in the text and listed at the end of the manuscript. The first seven references are presented correctly. However, some references are shown in the form of the author’s name and year of publication (for example: line 62, 94, 95, 98, 109, 133, 160, 162, 225, 262, 264, 265, 271, 276, 280, 323. Also, the sequence of presentation of some of these references in the list of references at the end of the manuscript is not in agreement with their appearance in the text. Among the errors present in references there are also no names of authors, no names of journal, incorrect abbreviated names of journals, no year of publication, no volume, and no pages of article. The authors should prepare references in accordance with instruction for authors and check their correctness using PubMed.
- Lines 66-68. The authors stated that “Damage to the pancreas in the course of diabetes mellitus occurs both in the form of dysfunction of the exocrine portion and progressive failure of endocrine function. Both abnormalities exacerbate the lack of control over normal blood glucose levels in the body” these sentences are unclear. How does “dysfunction of exocrine portion” “exacerbate the lack of control over normal blood glucose levels”?
- Line 75-78. The authors should add some words on diabetes mellitus type 3.
- Lines 80-81. Some sentences are broken and unclear: “In particular, these are patients with abnormal glycaemia [11].
- Line 82. What does “multinuclear leukocytes” mean? Probably, the authors wanted to write polymorphonuclear leukocyte. In addition, polymorphonuclear leukocytes exhibit the ability to phagocytosis. For this reason, the term “phagocytes“ in the fist part of the sentence should be replaced with “macrophages”.
- Line 129. Sars-CoV-2 means severe acute respiratory syndrome coronavirus 2 and for this reason the form “SARS-CoV-2 virus” is not correct.
- Line 153-155. Authors stated that “activating the Na+/H+ exchanger in the pancreas leads to hypoxia and cellular acidification, which damages pancreatic tissue”. Sodium/hydrogen exchanger transports Na+ into the cell and H+ out of the cell. How can it lead to cellular acidification?
- Line 167. What was a reason for introducing the abbreviation ARDS if SARS was already used?
- Line 231.
- Line 243. What severe insufficiency did the authors want to write about? In addition, glucocorticosteroids have possibly been used in severe cases of COVID-19. Therefore, it is difficult to say whether AP was due to glucocorticosteroids use or the primary severe course of COVID-19. It should be discussed in the manuscript.
- Lines 308-311. The form of the sentence is strange and should be corrected.
- The authors should present the interaction between coagulation and inflammation, as well as coagulation, CIVID-2, and pancreatitis. Coagulation stimulates the development of inflammation, and this relationship is two-sided. At the same time, inflammation activates the coagulation cascade (PMID: 19437587). COVID-19 predisposes patients to venous thromboembolism, due to excessive inflammation, platelet activation, and endothelial dysfunction (PMID: 32031570). Also, the development of acute pancreatitis is associated with activation of coagulation and the is a close relationship between degree of coagulation and the severity of acute pancreatitis. In mild acute pancreatitis, local scattered intravascular thrombosis within pancreatic microcirculation is observed. In severe acute pancreatitis, coagulation disorders may lead to development of disseminated intravascular coagulation (DIC) (PMID: 3515616; PMID: 12604906). Animal experimental (PMID: 18955758; PMID: 33077694; PMID: 32471279; PMID: 27754317; PMID: 26579579) and clinical (PMID: 19423455; PMID: 24944612; PMID: 30289392) studies indicate that inhibition of coagulation inhibits the development of AP and exhibits therapeutic effect in this disease. Similar therapeutic effect of heparin was found in COVID-19 (Di Micco et al; Viruses 2021, 13(12), 2486; doi.org/10.3390/v13122486). Heparin and derivatives are World Health Organization (WHO) recommended to prevent or treat thrombotic complications in moderate to severe COVID-19. Moreover, heparin can also inhibit viral adhesion to the cell membrane by interfering with heparan sulfate-dependent binding to angiotensin-converting enzyme 2 (ACE2) receptor (PMID: 34680499).
- The authors should also write that in the case of circulatory system disorders, blood is mainly directed to the heart and brain. The general regulation of blood flow dominates in the digestive system. Stress, trauma, hypoxemia, or severe infection lead to restriction of blood flow through the gastrointestinal tract. It should be expected therefore that in the case of COVID-19 there is a significant reduction in blood flow in the digestive system. On the other hand, the digestive system is especially sensitive to hypoxia and reduction in visceral blood flow. Blood flow adequate to needs of the organs plays a key role in maintaining their integrity. Clinical and experimental studies have showed that pancreatic ischemia may be the primary cause of acute pancreatitis and is involved in the progression of mild acute pancreatitis to the severe forms of the disease (PMID: 8831599; PMID: 10669996; PMID: 11453102). The early disturbance of pancreatic circulation is also observed in AP caused by other, primarily non-vascular mechanisms (PMID: 2089536). Moreover, previous studies indicated that the improvement of pancreatic blood flow inhibits the development of AP and accelerates the pancreatic recovery (PMID: 9444624; PMID: 14726612; PMID: 25594510). The above data indicate another mechanism of relationship between COVID-19 and the possibility of developing acute pancreatitis.
- Conclusions should be short and present only the main points of the article. The current version of the conclusion is almost three times longer than the abstract. It should be shortened.
Round 2
Reviewer 2 Report
Manuscript ID ijms-1499836, the second review.
Title: Consequences of COVID-19 for the pancreas
Authors: Abramczyk et al.
The new version of the manuscript “Consequences of COVID-19 for the pancreas” is better than previous one, but still contains some errors and deficiencies.
List of errors and deficiencies
- Previous comment 1. All abbreviations should be presented in their full name at the point where they appear for the first time in the abstract and repeated in the body of the manuscript (for example see line 8, 29, 51, 320, 321). On the other hand, the abundance of abbreviations makes the text difficult to understand (for example, line 75, EPI).
The authors replied that:” We explained the abbreviation. Abbreviation from line 29 was earlier explained in 26 line”.
Comment to the authors’ response: Full name of the abbreviation COVID-19 should be shown in line 8. Abbreviations, ACE2 and DPP4 shown in previous line 29 were not explained in any earlier lines in the first version of the manuscript. It is still unknown what means TMPRSS2 (line 32 in the second version of the manuscript). The authors’ responses should be accurate and factually correct!
- Previous comment 7. There is a total mess in references. References must be numbered in order of appearance in the text and listed at the end of the manuscript. The first seven references are presented correctly. However, some references are shown in the form of the author’s name and year of publication (for example: line 62, 94, 95, 98, 109, 133, 160, 162, 225, 262, 264, 265, 271, 276, 280, 323. Also, the sequence of presentation of some of these references in the list of references at the end of the manuscript is not in agreement with their appearance in the text. Among the errors present in references there are also no names of authors, no names of journal, incorrect abbreviated names of journals, no year of publication, no volume, and no pages of article. The authors should prepare references in accordance with instruction for authors and check their correctness using PubMed.
The authors replied that:” We Have corrected references in the text, and we corrected their list at the end of the manuscript”.
Comment to the authors’ response: There is still a total mess in references list. Some examples: In the reference 1, given names of authors is incomplete. The authors should write Zhang SF, Tuo JL, Huang XB etc. In addition, the authors provide pages/article number, e0191789. The reference 4, the authors should provide pages/article number, 59. The reference 7, it is unknown what is presented in this reference. It is not an article, maybe a book, but the authors should prepare all references in agreement with instructions for authors. The reference 9, page numbers are 1011-1017; doi is not correct. In addition, this reference is repeated as reference 44. The reference 11, the abbreviation of the journal name is Wiad Lek, whereas page numbers are 731-744. The reference 12, the abbreviation of the journal name is Infez Med. The reference 17. The name of the last author is not Prato S Del but Del Prato S. The article on COVID-19 was not published in 2017, but in 2020; volume number is not 8, but 9 and page numbers are 782-792. The reference 18, page numbers are 2211-2217. The reference 20, the page/article number is 2475. The reference 45, the authors should provide given names of authors in the form of initials. The authors should check correctness of all references using PubMed, as well as check whether sentences supported by references in the manuscript are in accordance with the content of these references.
- Previous comment 18. The authors should present the interaction between coagulation and inflammation, as well as coagulation, COVID-2, and pancreatitis…
The authors replied that Another mechanism of developing AP during COVID-19, may be a coagulation cascade activation caused by active inflammatory process due to SARS-CoV2 infection (PMID: 19437587). The ongoing inflammatory process causes not only hemostasis imbalance for blood clotting, but it also leads to intensification of coagulation by removing epithelial cell protein C receptor (EPCR) from epithelial by the means of inflammatory mediators and thrombin (PMID: 19437587) …”
Comment to the authors’ response: The changes made by the authors are generally correct. However, it is not possible to conduct a complete study with clinical observation for ethical reason. Therefore, some research must be carried out on animals. Such studies have provided hard evidence of the relationship between coagulation and acute pancreatitis. For this reason, it would be advisable in line 276, after reference 74, to insert the following sentence “These observations have been supported by the results of experimental studies showing that the inhibition of coagulation reduces the development of AP (PMID: 32471279; PMID: 27754317; PMID: 26579579) and exhibits therapeutic effect in this disease (PMID: 18955758; PMID: 33077694)”. Moreover, the authors should replace “depends” with “depend” (line 273/274) and replace “the most” with “more“ (the end of line 274).
- Previous comment 19. The authors should also write that in the case of circulatory system disorders, blood is mainly directed to the heart and brain. The general regulation of blood flow dominates in the digestive system. Stress, trauma, hypoxemia, or severe infection lead to restriction of blood flow through the gastrointestinal tract. It should be expected therefore that in the case of COVID-19 there is a significant reduction in blood flow in the digestive system. On the other hand, the digestive system is especially sensitive to hypoxia and reduction in visceral blood flow…
The authors replied that:” There is also a hypothesis, which claims that, AP can develop because of blood circulatory centralization resulting from uncontrolled cytokine storm created by SARS-CoV2 infection (PMID: 32592501). There exist reports, which say that pancreatic ischemia may be the cause of different degrees of acute pancreatitis(PMID: 8831599; PMID: 10669996). This statement can be supported by the reports which state that pancreatic blood reperfusion inhibits the development of AP and accelerate pancreas recovery(PMID: 9444624)”.
Comment to the authors’ response: The changes made by the authors are generally correct. It should be noted, however, that the last sentence concerns the protective and therapeutic effect of improving pancreatic blood flow in acute pancreatitis. The quoted article concerns only the protective effect. Therefore, it is advisable to quote all three suggested articles, including PMID: 14726612 and PMID: 25594510.
